# ZZEdit: ZigZag Trajectories of Inversion and Denoising for Zero-shot Image Editing

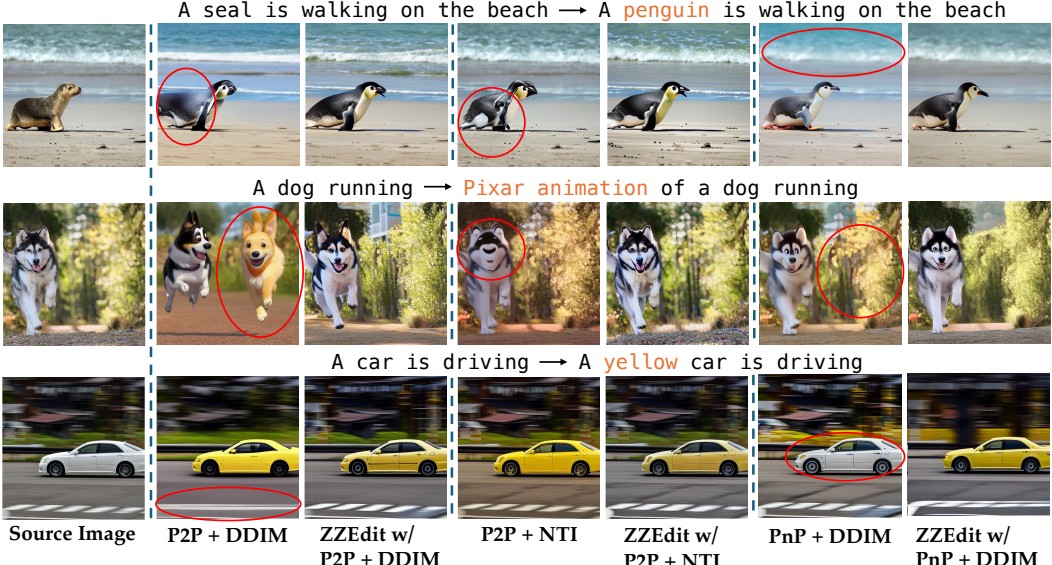

Figure 1: We propose a new editing paradigm dubbed ZZEdit, which demonstrates a more subtle editability and fidelity over the commonly employed "inversion-then-editing" pipeline. Moreover, it seamlessly integrates with contemporary text-driven image editing methods, such as P2P (Hertz et al., 2022) (with DDIM inversion (Song et al., 2020) or Null-text inversion (Mokady et al., 2023)) and PnP (Tumanyan et al., 2023) (with DDIM inversion), enhancing their capabilities. Without training or finetuning, our ZZEdit paradigm shows the feasibility of using dynamic latent trajectory on the existing image editing techniques.

## Abstract

Editability and fidelity are two essential demands for text-driven image editing, which expects that the editing area should align with the target prompt and the rest should remain unchanged separately. The current cutting-edge editing methods usually obey an "inversion-then-editing" pipeline, where the input image is first inverted to an approximate Gaussian noise $z_T$ with $T$ steps, based on which a sampling process is performed using the target prompt. Nevertheless, we argue that *it is not a good choice to use a near-Gaussian noise as a pivot for further editing since it almost lost all structure fidelity.* To verify this, we conduct a pilot experiment and find that the target prompt has different guiding degrees towards those latents on the inversion trajectory. Thus, a structure-preserving while sufficient-for-editing point is a more suitable pivot. Based on this, we propose a novel editing paradigm dubbed ZZEdit, which first locates such a pivot during the inversion trajectory and then mildly strengthens target guidance via the proposed ZigZag process. Concretely, our ZigZag process fulfills denoising and inversion iteratively, which gradually approaches the target while still holding background fidelity. Afterwards, to achieve the same number of inversion and denoising steps, we perform a pure sampling process under the target prompt. Extensive experiments highlight the effectiveness of our ZZEdit paradigm in diverse image editing scenarios compared with the existing "inversion-then-editing" pipeline.

# 1 INTRODUCTION

Recent years, large-scale text-guided diffusion models (Saharia et al., 2022; Rombach et al., 2022; Ramesh et al., 2022a; Yu et al., 2022; Gu et al., 2022) have attracted growing attention in computer vision and graphics community, showing efficiency for high-quality text-to-image (T2I) synthesis. To replicate this success into text-guided image editing and enable users to manipulate input images according to their text prompt, early attempts usually take additional user-provided masks (Gafni et al., 2022; Nichol et al., 2021; Ramesh et al., 2022b; Avrahami et al., 2023b; Mokady et al., 2022) or box (Li et al., 2023). Besides, (Zhang et al., 2023; Qin et al., 2023) take more conditions for fine-grained control over images e.g., depth maps, canny edges, poses, and sketches. Another line of research aims for *text-only* interactive image editing (Hertz et al., 2022; Tumanyan et al., 2023; Dong et al., 2023; Mokady et al., 2023; Cao et al., 2023; Ju et al., 2024; Bar-Tal et al., 2022; Meng et al., 2022). Since the last setting operates with minimal input conditions (i.e., only image and text) but also shows promising results for real image editing, we follow their trend in this work.

The current text-only image editing methods usually obey the "inversion-then-editing" pipeline. Specifically, as the reverse of DDIM sampling (Song et al., 2020), deterministic DDIM inversion gradually adds noise to the image feature $z_0$ for $T$ steps until it becomes an approximate Gaussian noise $z_T$. Then, Classifier-free Guidance (CFG) (Ho & Salimans, 2022) is applied for the sampling process, which denoises the inverted $z_T$ under the guidance of the target prompt. Here, we raise a question that *is it a good choice to directly invert the input image to a near-Gaussian noise*? To answer it, as seen in Fig. 2, we conduct a pilot experiment, which probes into the DDIM inversion, reconstruction, and editing process and discovers that the target prompt has different guidance degrees to those latents on the inversion/reconstruction trajectory. Thus, for the sake of both editability and fidelity, we argue that instead of using a near-Gaussian latent as a pivot, a more subtle way is to seek a point which keeps certain structure information while still having sufficient editability for subsequent denoising processes. From another perspective, we believe that different editing examples need to corrupt the input image to different degrees for subtle editing.

In view of the above considerations, this paper proposes a simple yet effective editing paradigm, dubbed ZZEdit, where the insight behind is *mildly strengthening guidance at a structure-preserving while sufficient-for-editing point*. Specifically, starting from $z_0$, we are constantly increasing the inversion degree, while looking for the *first* step on the inversion trajectory whose response to the target prompt is greater than that to the source one. We use this step as the editing *pivot* for our ZZEdit, which meets the requirements of maintaining structure and editability all at once. Then, as the core of ZZEdit, we propose a ZigZag process, which performs one-step editing and inversion alternately to mildly deepen the target editing from the selected pivot. Last, a pure editing process is conducted to ensure equal-step inversion and sampling. Note that our ZZEdit enjoys a unique advantage in maintaining vital structure (e.g., background) information while ensuring editability since we opt for a suitable intermediate inverted latent rather than a near-Gaussian latent, making it easier to alleviate the accumulated reconstruction error (Mokady et al., 2023; Dong et al., 2023).

As a novel editing paradigm, our ZZEdit can be applied to the existing inversion-based editing methods painlessly. Though simple, our ZZEdit is still principled, which casts a glance of applying dynamic trajectory of inversion and sampling to boost the existing editing methods. As shown in Fig. 1, we compare our ZZEdit with the typical "inversion-then-editing" pipeline using two methods P2P (Hertz et al., 2022) and PnP (Tumanyan et al., 2023). Specifically, P2P supports DDIM inversion and Null-Text inversion (NTI) (Mokady et al., 2023), in which the latter delivers better results by optimizing unconditional textual embeddings. However, P2P+NTI often leads to lower editability for shape editing (see the first row) or color leak (see the last row). Besides, PnP+DDIM sometimes yields background drift. In contrast, when our ZZEdit are equipped with these editing methods, more elegant editability and fidelity are achieved. To sum up, our main contributions are:

- We provide new empirical insights about locating a structure-preserving while sufficient-for-editing point during the inversion trajectory as the editing pivot.
- We propose a novel zero-shot image editing paradigm named ZZEdit, where a Zigzag process is designed to mildly enhance the target guidance at the suitable editing pivot.
- Extensive qualitative and quantitative experiments demonstrate that our ZZEdit is versatile across different editing methods, including P2P (Hertz et al., 2022) and PnP (Tumanyan et al., 2023), which achieves state-of-the-art editing performance.

## 2 RELATED WORKS

**Text-driven Image Generation.** Recent years, diffusion models (Song et al., 2020; Ho et al., 2020) has shown its capacity in text-to-image (T2I) generation. DALLE-2 (Ramesh et al., 2022a) proposes a two-stage model: a prior generating a CLIP (Radford et al., 2021) image embedding given a text caption, and a decoder producing an image conditioned on the image embedding. Building on the strength of diffusion models in high-fidelity image generation, Imagen (Saharia et al., 2022) discovers that large frozen language models trained only on text data are effective text encoders for text-to-image generation. Further, to enable diffusion models training on limited computational resources while retaining quality, Stable Diffusion (Rombach et al., 2022) trains models in the latent space of powerful pretrained autoencoders.

**Text-driven Image Editing.** Different from the general text-driven image generation, a group of methods turn the model for single-image editing. SDEdit (Meng et al., 2022) first adds noise to the input (e.g., stroke painting), then subsequently denoises through the prior from stochastic differential equation (SDE). DiffusionCLIP (Kim et al., 2022) proposes a text-guided image manipulation method using the pretrained diffusion models and CLIP loss. To further improve the editing fidelity, some approaches require a mask region (Avrahami et al., 2023a; 2022; Nichol et al., 2021), where the background out of the mask can remain the same while it can be time-consuming for users to provide a mask. Then, for text-only intuitive image editing, DiffEdit (Couairon et al., 2022) and MasaCtrl (Cao et al., 2023) automatically infer a mask according to the target prompt. P2P (Hertz et al., 2022) and PnP (Tumanyan et al., 2023) demonstrate that fine-grained control can be achieved by cross-attention layers and manipulating spatial features and their self-attention inside the model respectively. Besides, Imagic (Kawar et al., 2023) and UniTune (Valevski et al., 2022) conduct fine-tuning on Imagen (Saharia et al., 2022) to capture the image-specific appearance, which also does not need edit masks either. Further, Instructpix2pix (Brooks et al., 2023) proposes an editing method following instructions, where GPT-3 (Brown, 2020) and Stable Diffusion are used for dataset construction. Pix2Pix-Zero (Parmar et al., 2023) can perform image-to-image translation without manual prompting. Moreover, another line of techniques proposes to insert new concepts into a pretrained T2I model (e.g., a specified person, bag, cup) for personalize usage (Ruiz et al., 2023; Gal et al., 2022; Dong et al., 2022; Kumari et al., 2023; Smith et al., 2023; Tewel et al., 2023).

**Inversion in Editing Models.** The commonly-used DDIM inversion scheme (Song et al., 2020; Dhariwal & Nichol, 2021) conducts DDIM sampling in the reverse direction, which is effective for unconditional generation (reconstruction). However, when the classifier-free guidance (Ho & Salimans, 2022) is applied for editing purposes, the accumulated error of DDIM inversion would magnify and lead to a poor reconstruction, thus bringing unsatisfied editing results. To address this, several methods (Dong et al., 2023; Mokady et al., 2023) propose to perform optimization on inverted latents, where Null-text inversion (NTI) (Mokady et al., 2023) optimizes the unconditional textual embedding while Prompt-Tuning inversion (PTI) (Dong et al., 2023) optimizes the conditional embedding. There are also some techniques (Ju et al., 2024; Garibi et al., 2024; Wallace et al., 2023) improve DDIM inversion without fine-tuning.

Different from the above methods, this paper takes a close look at the latent trajectory of the existing "inversion-then-editing" pipeline, which usually takes an approximate-Gaussian latent as the editing pivot. However, we argue that it is usually a suboptimal solution. Without additional training or fine-tuning, we propose a new editing paradigm ZZEdit, which enhances editing mildly at a structure-preserving while sufficient-for-editing point, which considers structure and editability in the latent trajectory all at once.

## 3 PRELIMINARY

**Stable Diffusion (SD).** SD (Rombach et al., 2022) trains diffusion models for text-to-image generation in the latent space of an autoencoder $\mathcal{D}(\mathcal{E}(x))$. The encoder evaluates the latent feature $z = \mathcal{E}(x)$ for an input image while the decoder $\mathcal{D}$ maps the latent representation to the RGB space. In the forward process, the latent input $z_0$ is perturbed by Gaussian noise gradually, leading to $z_t$. To sequentially denoising, a U-Net (Ronneberger et al., 2015) $\epsilon_\theta$ containing a series of residual, self-attention, and cross-attention blocks is trained to predict the noise by the following loss objective:

$$\min_\theta E_{z_0, \epsilon \in (0,I), t \in \text{Uniform}(1,T)} \| \epsilon - \epsilon_\theta(z_t, t, \mathcal{C}) \|, \tag{1}$$

where $\mathcal{C}$ denotes the text embeddings. Once the network is trained, deterministic DDIM sampling (Song et al., 2020) can be applied to accurately reconstruct a given real image:

$$z_{t-1} = \sqrt{\frac{\alpha_{t-1}}{\alpha_t}} z_t + \left( \sqrt{\frac{1}{\alpha_{t-1}} - 1} - \sqrt{\frac{1}{\alpha_t} - 1} \right) \cdot \epsilon_\theta(z_t, t, \mathcal{C}). \tag{2}$$

**DDIM Inversion.** Previous editing techniques are always built upon text-guided diffusion models, e.g., SD. They use DDIM inversion (Song et al., 2020) to project an image into a known latent space before editing, which performs DDIM sampling process in a reverse way:

$$z_{t+1} = \sqrt{\frac{\alpha_{t+1}}{\alpha_t}} z_t + \left( \sqrt{\frac{1}{\alpha_{t+1}} - 1} - \sqrt{\frac{1}{\alpha_t} - 1} \right) \cdot \epsilon_\theta(z_t, t, \mathcal{C}). \tag{3}$$

The technique is based on the assumption that the ODE process can be reversed in the limit of small steps. Here, a ideal inversion trajectory from $z_{t-1}$ to $z_t$ should align with the direction from $z_t$ to $z_{t-1}$ in denoising trajectory.

**Classifier-free Guidance (CFG).** To enhance the guidance of the text guidance in text-guided generation, classifier-free guidance (Ho & Salimans, 2022) is proposed, where both conditioned prediction and unconditioned prediction are performed at each step. The calculation is defined as:

$$\tilde{\epsilon}_\theta(z_t, t, \mathcal{C}, \varnothing) = \omega \cdot \epsilon_\theta(z_t, t, \mathcal{C}) + (1 - \omega) \cdot \epsilon_\theta(z_t, t, \varnothing), \tag{4}$$

where $\varnothing$ is the embeddings of a null text, and $\omega$ is the guidance scale parameter. Note that DDIM inversion can nearly reconstruct the original image when $\omega = 1$ (Mokady et al., 2023), where a slight error is introduced in each step. However, a large guidance scale $\omega > 1$ is necessary for the editing task, which would magnify such accumulated error.

## 4 METHODS

Given an input image $I$ and a target prompt $\mathcal{P}_{tgt}$, text-driven image editing tries to achieve two fundamental needs: ***editability*** and ***fidelity***. The former aims to change visual content to be consistent with the textual description of $\mathcal{P}_{tgt}$, while the latter requires the rest to remain unchanged. In this section, we first analyze the existing "inversion-then-editing" pipeline in Sec. 4.1, which shows the target prompt $\mathcal{P}_{tgt}$ has distinct guidance degrees to those latents on the inversion trajectory. Then, the overview of the proposed ZZEdit is given in Sec. 4.2, which first locates a structure-preserving while sufficient-for-editing point as the editing pivot and then performs a mild target guiding process still holding structure information based on this pivot. We elaborate on these two parts in Sec. 4.3 and Sec. 4.4. Our ZZEdit paradigm can be applied painlessly in those methods which obey "inversion-then-editing" and improve their performance.

### 4.1 PILOT ANALYSIS OF THE TYPICAL "INVERSION-THEN-EDITING" PIPELINE

Recent text-only image editing pipeline always directly invert the input image $I$ for $T$ steps to obtain an approximately standard Gaussian noise $z_T$, from which an edited image is sampled under the guidance of target prompt $\mathcal{P}_{tgt}$ using CFG. However, *is it a good choice to directly invert the input image to a near-Gaussian noise*? In this paper, we argue the answer is negative. Next, we will conduct a pilot experiment to explore this with commonly-used DDIM inversion.

In Fig.2, we divide the T-step process into five parts evenly. Recall that DDIM inversion can nearly reconstruct the original image $z_0$ when CFG scale $\omega = 1$, the reconstructed latent $\hat{z}_t$ is approximately equal to the inverted latent $z_t$ (i.e., $z_t \approx \hat{z}_t$). Then, we attempt to quantitatively measure the target guidance degree for the latent $\hat{z}_t$ in reconstruction trajectory, where $t \in [\frac{1}{5}T, \frac{2}{5}T, \frac{3}{5}T, \frac{4}{5}T, T]$. Specifically, we use CFG scale $\omega = 7.5$ for editing, where DDIM sampling is conducted on different reconstructed $\hat{z}_t$ for $\frac{1}{5}T$ steps, yielding $\tilde{z}_{t-\frac{1}{5}T}$ as $\tilde{z}_0$, $\tilde{z}_{\frac{1}{5}T}$, $\tilde{z}_{\frac{2}{5}T}$, $\tilde{z}_{\frac{3}{5}T}$, and $\tilde{z}_{\frac{4}{5}T}$, respectively. Next, we use superscript to distinguish different editing trajectories: ①$\hat{z}_{\frac{1}{5}T} \to \tilde{z}_0^1$, ②$\hat{z}_{\frac{2}{5}T} \to \tilde{z}_{\frac{1}{5}T}^2 \to \tilde{z}_0^2$, ③$\hat{z}_{\frac{3}{5}T} \to \tilde{z}_{\frac{2}{5}T}^3 \to \tilde{z}_{\frac{1}{5}T}^3 \to \tilde{z}_0^3$, ④$\hat{z}_{\frac{4}{5}T} \to \tilde{z}_{\frac{3}{5}T}^4 \to \tilde{z}_{\frac{2}{5}T}^4 \to \tilde{z}_{\frac{1}{5}T}^4 \to \tilde{z}_0^4$, and ⑤$\hat{z}_T \to \tilde{z}_{\frac{4}{5}T}^5 \to \tilde{z}_{\frac{3}{5}T}^5 \to \tilde{z}_{\frac{2}{5}T}^5 \to \tilde{z}_{\frac{1}{5}T}^5 \to \tilde{z}_0^5$. Such superscript is omitted in Fig.2 for simplicity.

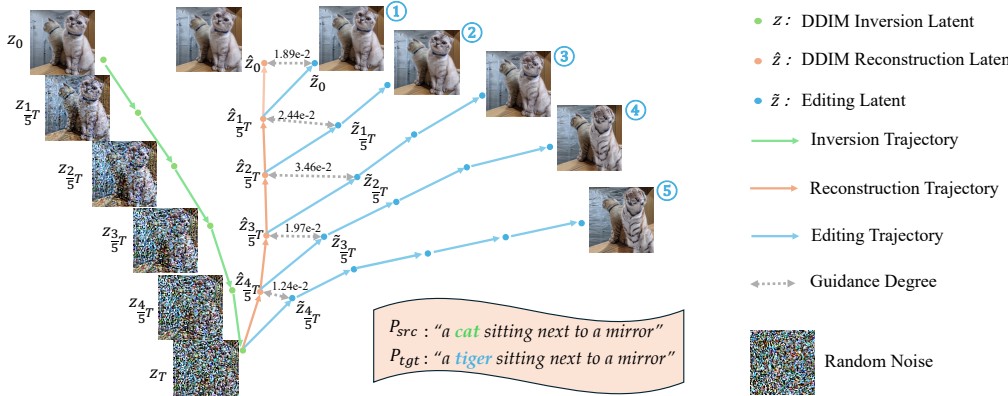

Figure 2: Pilot experiment on the typical "inversion-then-editing" pipeline, which takes source prompt $\mathcal{P}_{src}$, target prompt $\mathcal{P}_{tgt}$, and a clean image latent $z_0$ as input. In the DDIM inversion trajectory, $z_0$ is inverted into a near-Gaussian latent $z_T$ using source prompt $\mathcal{P}_{src}$ and UNet $\epsilon_\theta$. At the reconstruction stage of DDIM denoising, $\hat{z}_0$ can nearly reconstruct $z_0$ under $T$ steps when the CFG scale $\omega = 1$. At the editing stage of the DDIM denoising, the target prompt $\mathcal{P}_{tgt}$ has distinct guidance degrees to the reconstructed latent $\hat{z}_t$ on different step $t$.

In this way, we can measure the guidance degree of the target prompt $\mathcal{P}_{tgt}$ to the reconstructed $\hat{z}_t$ by calculating the norm of difference between $\hat{z}_t$ and $\tilde{z}_t$ (i.e., $\|\tilde{z}_t - \hat{z}_t\|$). For instance, the largest guidance degree occurs in the editing trajectory of ③ in our pilot analysis. Since $z_t \approx \hat{z}_t$, a general discovery can be obtained: the target prompt $\mathcal{P}_{tgt}$ has different guidance degrees to those latents on the inversion trajectory and the maximum one is not necessarily at the $T$ step.

## 4.2 OVERVIEW OF THE PROPOSED ZZEDIT

Although the editability of different latent on inversion trajectory is distinct, the existing "inversion-then-editing" pipeline directly inverts the input image to a near-Gaussian latent, which almost loses all structure fidelity. As contrast, this paper seeks a structure-preserving while sufficient-for-editing point for the subsequent target guidance. Specifically, as seen in the left part of Fig. 3, we propose a new editing paradigm, dubbed ZZEdit, which can be divided into three steps:

(i) We first locate a suitable point in inversion trajectory as an editing pivot, which is with certain structure information while still sufficient for editing, which is elaborated in Sec.4.3.

(ii) A mild guiding process, named ZigZag process is conducted to enhance target guidance at the pivot. Specifically, it consists of alternately performing one-step denoising and one-step inversion, which is detailed in Sec.4.4.

(iii) The remaining comprises a diffusion process guided by the target prompt to achieve equal-step inversion and sampling. Note that when equipping the existing editing method with our ZZEdit, the denoising process needs to retain the characteristics of the method, such as P2P (Hertz et al., 2022) injecting cross-attention maps and PnP (Tumanyan et al., 2023) injecting self-attention maps. We summarize applying our ZZEdit to the existing text-driven image editing methods in Alg. 1.

## 4.3 LOCATING A SUITBALE PIVOT ON THE INVERSION TRAJECTORY

We consider editability by seeking a point on inversion trajectory which has a larger response towards the target prompt $\mathcal{P}_{tgt}$ than source prompt $\mathcal{P}_{src}$. Specifically, starting from $z_0$, we gradually increase the inversion degree, leading to a series of inverted latent $\{z_0, z_1, ..., z_T\}$. As seen the right part of Fig. 3, we can obtain the reconstructed latent $\hat{z}_t$ using source prompt $\mathcal{P}_{src}$, the edited latent $\tilde{z}_t$ using target prompt $\mathcal{P}_{tgt}$, the reconstructed latent $\bar{z}_t$ using null text $\varnothing$ by DDIM sampling (Eq. 2). Here, we leverage the output of the denoising UNet $\epsilon_\theta$ to indicate the response level of different inverted latent $z_t$ to $\mathcal{P}_{src}$, $\mathcal{P}_{tgt}$, and $\varnothing$:

$$\epsilon_{t-1}^{src} \leftarrow \epsilon_\theta(z_t, t, \mathcal{C}_{src}), \quad \epsilon_{t-1}^{tgt} \leftarrow \epsilon_\theta(z_t, t, \mathcal{C}_{tgt}), \quad \epsilon_{t-1}^{\varnothing} \leftarrow \epsilon_\theta(z_t, t, \varnothing), \tag{5}$$

where $\mathcal{C}_{src}$ and $\mathcal{C}_{pgt}$ denotes the text embeddings of $\mathcal{P}_{src}$ and $\mathcal{P}_{src}$ separately.

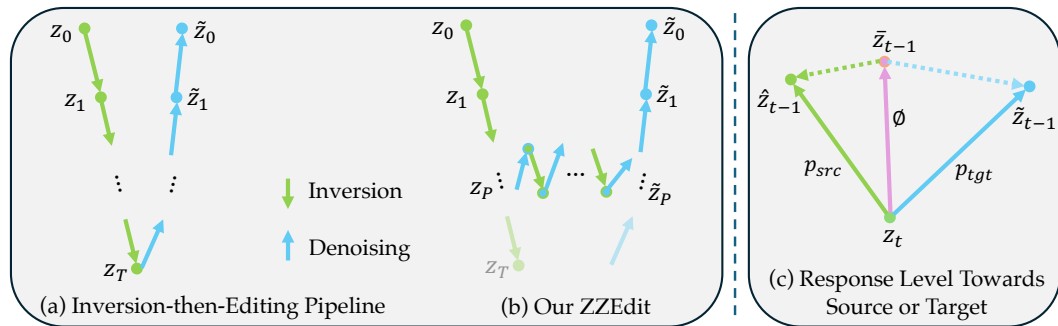

(a) Inversion-then-Editing Pipeline   (b) Our ZZEdit   (c) Response Level Towards Source or Target

Figure 3: Left: (a) The trajectory of the existing "inversion-then-editing" pipeline, which invertes the input image $z_0$ with $T$ steps using the source prompt $\mathcal{P}_{src}$, and then performs denoising under the target prompt $\mathcal{P}_{tgt}$. (b) The trajectory of our ZZEdit, where we first locate a structure-preserving while sufficient-for-editing point (marked as $P$) on the inversion trajectory, and then use a ZigZag process to mildly guide the latent towards the target. Afterwards, a pure denoising process is leveraged for the equal step of inversion and denoising. Right: (c) Illustration of the response of the inverted latent $z_t$ towards $\mathcal{P}_{src}$, null text $\varnothing$, and $\mathcal{P}_{tgt}$, which are represented with dotted lines. The more the response deviates from that of null text $\varnothing$, the greater the response.

Generally speaking, the latent $z_t$ with low-degree inversion would be more responsive to source prompt $\mathcal{P}_{src}$ due to limited corruption. As the inversion deepens, we can easily find those points whose response to target prompt $\mathcal{P}_{tgt}$ is greater than that to source prompt $\mathcal{P}_{src}$, where $\|\epsilon_{t-1}^{tgt} - \epsilon_{t-1}^{\varnothing}\| > \|\epsilon_{t-1}^{src} - \epsilon_{t-1}^{\varnothing}\|$. That is, the more the response deviates from that of null text $\varnothing$, the greater the response. To keep both editability and fidelity, we only locate the *first* point during inversion which has a larger target response as our editing pivot. We mark the satisfied step $t$ as $P \in [0, 1, ..., T]$. We show the pseudo algorithm in part I of Alg. 1.

## 4.4 MILD GUIDING: ZIGZAG PROCESS

To mildly deepen target guidance without ruining structure fidelity, we propose a ZigZag process based on the located pivot, which alternately execute single-step sampling and inversion.

**Mild Guiding.** As illustrated in Fig.3, the proposed ZigZag process is started after a $P$-step inversion. Formally, a full Zigzag process includes $K$ denoising steps and $K$ inversion steps, which are conducted alternately. We treat a denoising step and inversion step as a union, thus the ZigZag process consists of $K$ unions. Here, the inversion of $k$-th union can be expressed as:

$$z_t^k = \sqrt{\frac{\alpha_t}{\alpha_{t-1}}} z_{t-1}^k + \left( \sqrt{\frac{1}{\alpha_t} - 1} - \sqrt{\frac{1}{\alpha_{t-1}} - 1} \right) \cdot \epsilon_\theta(z_{t-1}^k, t, \mathcal{C}_{src}), \quad (6)$$

where $k \in \{1, 2, ..., K\}$. Then, the denoising step of $(k+1)$-th union in ZigZag process is:

$$z_{t-1}^{k+1} = \sqrt{\frac{\alpha_{t-1}}{\alpha_t}} z_t^k + \left( \sqrt{\frac{1}{\alpha_{t-1}} - 1} - \sqrt{\frac{1}{\alpha_t} - 1} \right) \cdot \epsilon_\theta(z_t^k, t, \mathcal{C}_{tgt}), \quad (7)$$

Substituting Eqn. 6 into Eqn. 7, we can obtain:

$$z_{t-1}^{k+1} = z_{t-1}^k + \left( \sqrt{\frac{1}{\alpha_{t-1}} - 1} - \sqrt{\frac{1}{\alpha_t} - 1} \right) \cdot \left( \epsilon_\theta(z_t^k, t, \mathcal{C}_{tgt}) - \epsilon_\theta(z_{t-1}^k, t-1, \mathcal{C}_{src}) \right), \quad (8)$$

where $\left( \sqrt{\frac{1}{\alpha_{t-1}} - 1} - \sqrt{\frac{1}{\alpha_t} - 1} \right) > 0$ according to noise schedule of diffusion models (Song et al., 2020). Thus, compared with the inverted latent in $k$-th union (i.e., $z_{t-1}^k$), the denoising latent in $(k+1)$-th union (i.e., $z_{t-1}^{k+1}$) would move towards target direction.

**Zigzag Steps.** For a fair comparison, we use the same steps of inversion and sampling with the typical "inversion-then-editing" pipeline to determine the number of ZigZag steps. That is, $P + K =$

---

**Algorithm 1:** ZZEdit: ZigZag Trajectories for Zero-shot Image Editing

---

**Input:** The inverted latents $\{z_0, z_1, ..., z_T\}$, source prompt $\mathcal{P}_{src}$, and target prompt $\mathcal{P}_{tgt}$
**Output:** An edited image or latent embedding $\tilde{z}_0$

---

**1** Part I: Locating a Structure-preserving While Sufficient-for-editing Step $P$ as the editing pivot

**2** **for** $t = 0 \rightarrow T$ **do**
**3**     $\epsilon_{t-1}^{src} \leftarrow \epsilon_\theta(z_t, t, \mathcal{C}_{src})$;    // caculate $\epsilon_{t-1}^{src}$ conditioning on $\mathcal{P}_{src}$
**4**     $\epsilon_{t-1}^{tgt} \leftarrow \epsilon_\theta(z_t, t, \mathcal{C}_{tgt})$;    // caculate $\epsilon_{t-1}^{tgt}$ conditioning on $\mathcal{P}_{tgt}$
**5**     $\epsilon_{t-1}^{\varnothing} \leftarrow \epsilon_\theta(z_t, t, \varnothing)$;    // caculate $\epsilon_{t-1}^{\varnothing}$ conditioning on $\varnothing$
**6**     **if** $\|\epsilon_{t-1}^{tgt} - \epsilon_{t-1}^{\varnothing}\| > \|\epsilon_{t-1}^{src} - \epsilon_{t-1}^{\varnothing}\|$ **then**
**7**        break
**8**     **end**
**9** **end**
**10** **Return** $t$

---

**11** Part II: ZigZag Process

---

**12** **for** $t = P$ **do**
**13**     ZigZag Process alternately executs one-step denoising (Eq. 7) and inversion (Eq. 6);
**14** **end**

---

**15** Part III: Continuous Denoising Process

---

**16** **for** $t = P \rightarrow 0$ **do**
**17**     Denoising step equiped with exsiting image editing techniques such as P2P and PnP;
**18** **end**

---

$T$. Then, when the located editing pivot reaches $T$ steps (i.e., $P = T$), we make ZZEdit degenerate to the typical "inversion-then-editing" pipeline. Besides, to flexibly control the number of ZigZag steps, we additionally introduce a hyper-parameter $a$ as:

$$K = a \cdot (T - P), \tag{9}$$

where $a \in [0, 1]$. When $a = 0$, a continuous $P$-step sampling is performed from the located editing pivot without the ZigZag process. When $a = 1$, our ZZEdit realizes $T$ inversion and sampling steps separately, consuming the same UNet operations as the typical "inversion-then-editing" pipeline.

## 5 EXPERIMENT

### 5.1 EXPERIMENTAL SETUP

**Implementation Details.** We perform all experiments on a single Tesla A100 GPU using Py-Torch (Paszke et al., 2019). Following (Tumanyan et al., 2023), we use 50 steps as the DDIM schedule and the classifier-free guidance of 7.5 for editing. Besides, we use the official code of Stable Diffusion with version 1.5. For a fair comparison, we adopt the same cross-attention injection parameters and self-attention injection parameters when our ZZEdit is applied in the editing methods P2P (Hertz et al., 2022) and PnP (Tumanyan et al., 2023). In practice, to save time and computation, when looking for the editing pivot, we can only search from $[0.4T, 0.5T, ..., 0.9T, T]$, rather than $[0, 1, ..., T]$. The reasons here are: (1) low-degree inversion generally struggles to bring sufficient editability and (2) there is no need to look up each step.

**Evaluation Metrics.** To illustrate the effectiveness of our proposed ZZEdit, we use images from the dataset PIE-Bench (Ju et al., 2024). The editing results are evaluated on three aspects: structure distance (Tumanyan et al., 2022), background preservation covering PSNR (Huynh-Thu & Ghanbari, 2012), SSIM (Wang et al., 2004), MSE, and LPIPS (Zhang et al., 2018), and editing consistency of the whole image and regions in the editing mask, denoted as CLIP similarity (Wu et al., 2021).

### 5.2 ABLATION STUDIES

We ablate several key designs of our ZZEdit paradigm, which aims to answer the following questions. **Q1:** What is the difference between using different points on the inversion trajectory as the

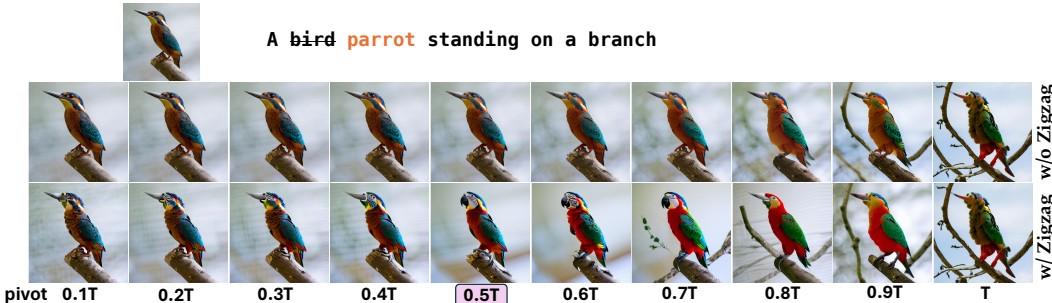

Figure 4: Ablation study of ZZEdit on P2P (Hertz et al., 2022) *w/* DDIM inversion. The first row displays the editing results when different points on the inversion trajectory are leveraged as the editing pivot, where no ZigZag process is equipped. Then, the second row shows the performance of using the ZigZag process additionally. Our method first locates a suitable pivot (marked with purple) and then mildly performs target guiding, which yields the most elegant editing results.

Table 1: Quantitative ablation study on the proposed ZigZag process with P2P (Hertz et al., 2022) *w/* DDIM inversion, where measurement results are obtained on the PIE-Bench dataset (Ju et al., 2024). We mark the best results in the ZigZag process in bold. We also provide the performance of selecting the editing pivot randomly, where the ZigZag process is equipped.

| Method | | Structure | Background Preservation | | | | CLIP Similariy | |
|---|---|---|---|---|---|---|---|---|
| | | L2 ↓ | PSNR↑ | LPIPS ↓ | MSE ↓ | SSIM ↑ | Whole↑ | Edited↑ |
| **P2P+DDIM Baseline** | | 69.41 | 17.88 | 208.37 | 219.11 | 71.30 | 25.01 | 22.44 |
| *w/* **Pivot** | *w/o* **ZigZag** ($a = 0$) | **22.60** | **23.71** | **107.01** | **68.27** | **79.60** | 24.43 | 21.52 |
| | *w/* **ZigZag** ($a = 0.2$) | 27.50 | 22.97 | 116.02 | 82.79 | 78.71 | 24.70 | 22.04 |
| | *w/* **ZigZag** ($a = 0.6$) | 28.26 | 22.48 | 122.36 | 87.26 | 77.94 | 25.07 | 22.14 |
| | *w/* **ZigZag** ($a = 1$) | 31.99 | 21.92 | 131.57 | 96.95 | 76.98 | **25.29** | **22.47** |
| **Random Pivot** *w/* **ZigZag** ($a = 1$) | | 25.84 | 24.07 | 105.36 | 81.43 | 79.56 | 24.76 | 21.84 |

editing pivot? **Q2:** Could the proposed ZigZag process enhance the target guidance at the suitable pivot? **Q3:** Could our ZZEdit locate a suitable pivot, which maintains both structure and editability?

**Different Editing Pivot in ZZEdit.** In Fig. 4, we answer the first question by applying our ZZEdit on P2P (Hertz et al., 2022) *w/* DDIM inversion and report the performance of selecting an editing pivot from $[0.1T, 0.2T, ..., 0.9T, T]$ during inversion trajectory. The first row uses $P$-step inversion and $P$-step sampling without the ZigZag process. The second row displays the results of ZZEdit using different editing pivots with the ZigZag process, where each result satisfies $P + K = T$.

From the first row, we can observe that the target prompt has different guiding abilities to different points on the inversion trajectory. For example, when we choose $[0.1T, 0.2T, 0.3T, 0.4T]$ as the editing pivot, structure fidelity is maintained well, but editability is poor. It demonstrates that a low-degree inversion struggles to bring sufficient editability. Besides, we notice that when using high-degree invertion (e.g.,$[0.8T, 0.9T, T]$), the results deliver satisfactory editability but an unpleasing background fidelity due to the amplification of accumulated errors. As for the second row, we equip the proposed ZigZag process at different editing pivots. Note that for those low-degree inversion latents, using the ZigZag process for target guidance also shows limited editability. Fortunately, our method first finds a structure-preserving while sufficient-for-editing point (marked in purple), and then performs mild guiding with the ZigZag process, which yields the best editing performance. We also use GPT-4V(ision) system (OpenAI, 2023) to evaluate Fig. 4 in Appendix.

**The Effectiveness of The ZigZag Process.** We answer the second question by employing different ZigZag steps on a suitable editing pivot, which makes $a$ in Eq. 9 take the value from $\{0, 0.2, 0.6, 1\}$. Fig. 5 shows a qualitative comparison on different baselines. Our ZZEdit can mildly approach the editing purpose through the increasing number of ZigZag steps while holding a satisfying background. We also provide a quantitative experiment in Tab. 1. When no ZigZag steps are employed ($a = 0$), the best background and structure can be obtained. However, it cannot achieve pleasing editing consistency. Besides, when we equip the ZigZag process, the gradual increase of ZigZag steps ($a = 0.2, 0.6$, and 1) can improve editing consistency. The quantitative ablation on the ZigZag

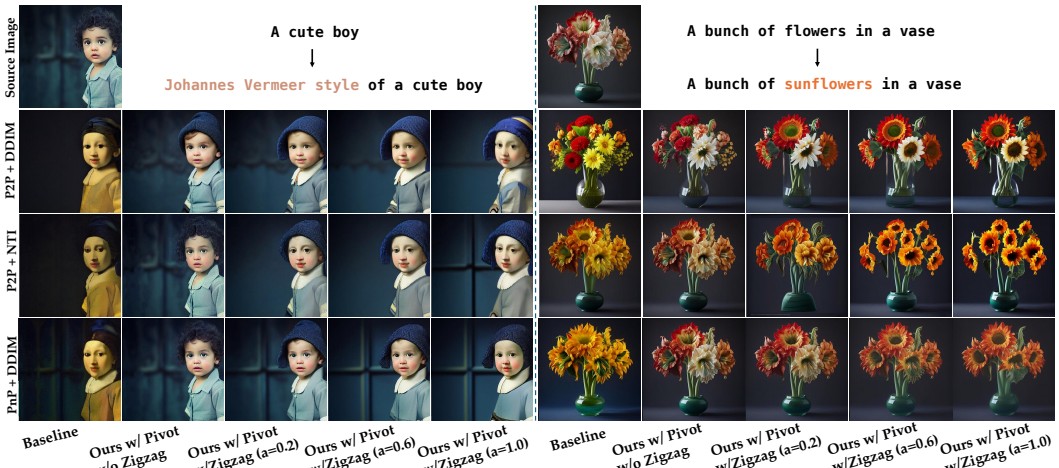

Figure 5: Qualitative ablation study on our ZigZag process with P2P (Hertz et al., 2022) and PnP (Tumanyan et al., 2023), which can mildly enhance the target guidance at a suitable pivot.

process with P2P *w/* Null-text inversion (NTI) and PnP *w/* DDIM inversion can be found in our Appendix.

**The Effectiveness and Distribution of Our Located Pivot.** To prove the effectiveness of the located pivot of our ZZEdit, we compare with the results of selecting an editing pivot randomly from $[0.1T, 0.2T, ..., 0.9T, T]$, where the ZigZag process ($a = 1$) is leveraged. Compared with the P2P baseline, although "random pivot *w/* ZigZag" can achieve excellent background and structure preservation, but its editing consistency is poor. Such poor editability of the random pivot proves the effectiveness of the editing pivot located in our ZZEdit. Besides, as seen in Fig. 6, we provide the distribution of the suitable editing pivot in our ZZEdit on the PIE-Bench dataset (Ju et al., 2024). Note that to save time and computation, we only look for the pivot from $[0.4T, 0.5T, ...0.9T, T]$ in practice. When the pivot reaches $T$ (i.e., $P = T$), our ZZEdit degenerates into the typical "inversion-then-editing" pipeline.

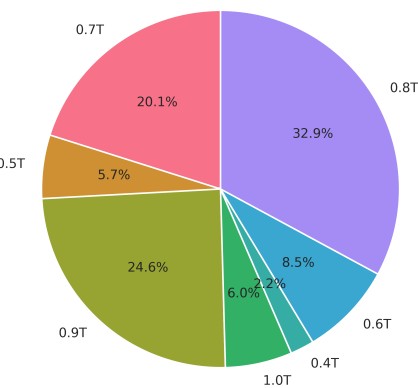

Figure 6: The statistics on the editing pivot located by our ZZEdit on the PIE-Bench dataset.

### 5.3 QUANTITATIVE RESULTS

To prove the superiority of the proposed ZZEdit, we compare it with the results of P2P (Hertz et al., 2022) and PnP (Tumanyan et al., 2023) under different inversion settings. As seen in Table. 2, when applying ZZEdit to P2P or PnP *w/* DDIM inversion, background, structure, and editing consistency are boosted steadily. Besides, although our ZZEdit does not yield better background and structure preservation compared with P2P *w/* NTI (or P2P *w/* Pnp inversion), it achieves a much higher editing consistency. We argue that such a trade-off is worthwhile for the image editing task.

### 5.4 QUALITATIVE RESULTS

In Fig. 7, we show a qualitative comparison with the current editing methods, including P2P (Hertz et al., 2022) *w/* DDIM inversion or NTI, PnP (Tumanyan et al., 2023) *w/* DDIM inversion, Pix2Pix-Zero (Parmar et al., 2023), Instructpix2pix (Brooks et al., 2023), and Masactrl (Cao et al., 2023). The editing scenario here includes attribute editing, object replacement, style transfer and background editing. Our ZZEdit paradigm can consistently improve the performance of P2P and PnP. Compared with other state-of-the-art methods, our ZZEdit shows its superiority through better background fidelity and editing consistency. More comparisons of editing results can be found in the Appendix.

Table 2: Comparison between ZZEdit and "inversion-then-editing" pipeline on P2P (Hertz et al., 2022) and PnP (Tumanyan et al., 2023) under different inversion settings: DDIM Song et al. (2020), NTI (Mokady et al., 2023), PTI (Dong et al., 2023), and Pnp inversion (Ju et al., 2024).

| Method | | Structure | Background Preservation | | | | CLIP Similariy | |
|---|---|---|---|---|---|---|---|---|
| Editing | Inv Setting | L2 ↓ | PSNR↑ | LPIPS ↓ | MSE ↓ | SSIM ↑ | Whole↑ | Edited↑ |
| P2P | DDIM | 69.41 | 17.88 | 208.37 | 219.11 | 71.30 | 25.01 | 22.44 |
| | NTI | 13.72 | 27.05 | 60.74 | 35.89 | 84.27 | 24.75 | 21.86 |
| | PTI | 16.17 | 26.21 | 69.01 | 39.73 | 83.40 | 24.61 | 21.87 |
| | Pnp_inv | **11.65** | **27.22** | **54.55** | **32.86** | **84.76** | 25.02 | 22.10 |
| | ZZEdit (*w/* DDIM) | 31.99 | 21.92 | 131.57 | 96.95 | 76.98 | **25.29** | **22.47** |
| | ZZEdit (*w/* NTI) | 16.15 | 25.67 | 84.28 | 49.06 | 82.14 | 25.16 | 22.13 |
| PnP | DDIM | 28.22 | 22.28 | 113.46 | 83.64 | 79.05 | 25.41 | 22.55 |
| | Pnp_inv | 24.29 | 22.46 | 106.06 | 80.45 | 79.68 | 25.41 | 22.62 |
| | ZZEdit (*w/* DDIM) | **23.49** | **24.55** | **86.61** | **55.04** | **82.18** | **25.43** | **22.91** |

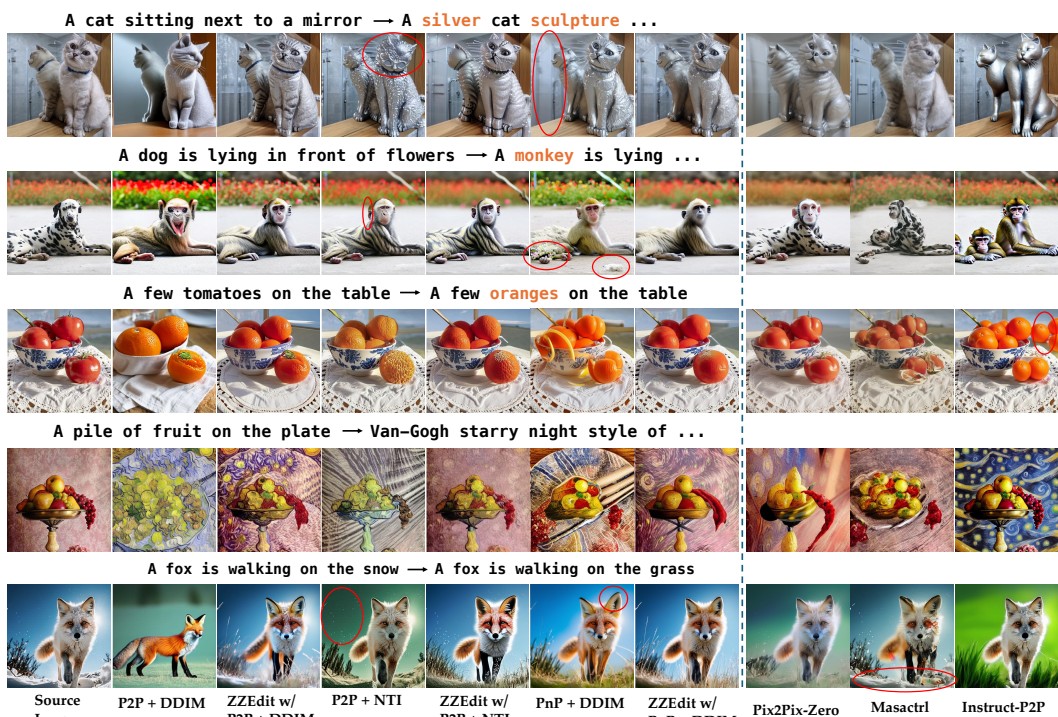

Figure 7: Visualization results of different editing techniques. From left to right: source image, P2P (Hertz et al., 2022) *w/* DDIM inversion, our ZZEdit applied on P2P *w/* DDIM inversion, P2P *w/* Null-text inversion, our ZZEdit applied on P2P *w/* Null-text inversion, PnP (Tumanyan et al., 2023) *w/* DDIM inversion, our ZZEdit applied on PnP *w/* DDIM inversion, Pix2Pix-Zero (Parmar et al., 2023), Masactrl (Cao et al., 2023), Instructpix2pix (Brooks et al., 2023).

# 6 CONCLUSION

We presented a novel zero-shot image editing paradigm, dubbed ZZEdit. Based on the observation that different latents on the inversion trajectory have distinct response levels to the target prompt, we proposed to select a point maintaining both structure information and sufficient editability as the editing pivot. Then, a ZigZag process was designed to execute sampling and inversion alternately, which mildly approach the target. Finally, we conducted a sampling process solely to keep the same-step inversion and sampling. Comprehensive experiments have shown that our method achieves outstanding outcomes across a broad spectrum of text-driven image editing methods.

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
