# APPENDIX FOR "ZZEDIT: ZIGZAG TRAJECTORIES OF INVERSION AND DENOISING FOR ZERO-SHOT IMAGE EDITING"

This Appendix includes 3 sections. Sec. 1 gives more ablation study results. Sec. 2 illustrates more qualitative results to compare our results with state-of-the-art image editing methods. Sec. 3 introduces the limitations and future work of our ZZEdit.

## 1 MORE ABLATION STUDY

**Different Editing Pivot in ZZEdit.** Recall that we provide the visualization results of using different points on the inversion trajectory as the editing pivot in Fig. 4 in our main paper. Here, we display one more visualization example in A-Fig. 1. Here, we mark our located editing pivot with purple. Although the background corresponding to low-degree inversion is well maintained, its editability is insufficient. In contrast, a high-degree inversion brings editability but loses fidelity gradually. To better evaluate the effect of different editing pivots, as shown in A-Fig. 3 and A-Fig. 4, we leverage GPT-4V(ision) system (OpenAI, 2023), which gives the editing comments by a Multimodal LLMs.

**The Effectiveness of The ZigZag Process.** We evaluate the effect of the proposed ZigZag process quantitatively based on the P2P (Hertz et al., 2022) *w/* DDIM inversion in Tab. 1 of our main paper. As seen in A-Tab. 1, we additionally provide the corresponding quantitative ablation results using PnP (Tumanyan et al., 2023) *w/* DDIM inversion and P2P *w/* Null-text inversion (Mokady et al., 2023). With the increase of $a$, our proposed Zigzag process gradually increases editing consistency, thus obtaining better CLIP similarity. While editing consistency increases, the performance of background preservation and structural information is slightly weakened.

**The Effectiveness of Our Located Pivot.** In A-Tab. 1, we also report the performance of selecting editing pivot from $[0.1T, 0.2T, ...0.9T, T]$ randomly, where the standard ZigZag process ($a = 1$) is equipped. It delivers excellent background and structure preservation, but very poor editability. This also demonstrates the efficiency of our located pivot.

## 2 MORE IMAGE EDITING RESULTS

As shown in A-Fig. 2, we show more qualitative comparison with the current text-driven editing methods, including P2P (Hertz et al., 2022) *w/* DDIM inversion and *w/* Null-text inversion,

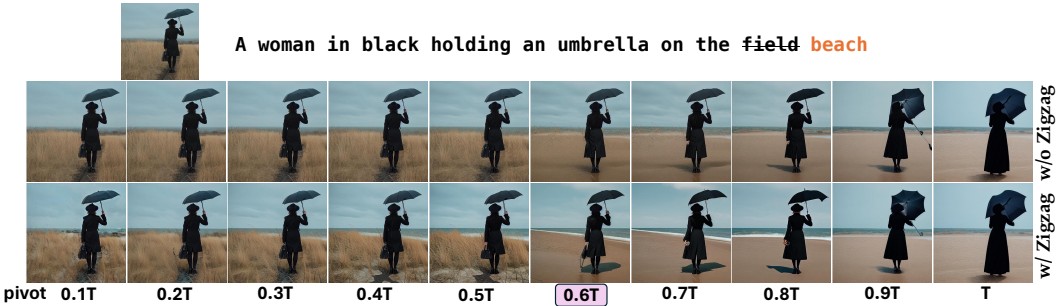

A woman in black holding an umbrella on the ~~field~~ beach

**A-Fig.** 1: More ablation results of applying ZZEdit on P2P (Hertz et al., 2022) *w/* DDIM inversion, where different inverted latents are used with or without the ZigZag process equipped.

**A-Tab.** 1: Quantitative ablation study on the proposed ZigZag process with PnP (Tumanyan et al., 2023) *w/* DDIM inversion and P2P (Hertz et al., 2022) *w/* Null-text inversion. Results are obtained on the PIE-Bench dataset (Ju et al., 2024). The best results in the ZigZag process are marked in bold. Here, the results of random pivot with the ZigZag process are also provided.

| Method | | Structure | Background Preservation | | | | CLIP Similariy | |
|---|---|---|---|---|---|---|---|---|
| | | L2 ↓ | PSNR↑ | LPIPS ↓ | MSE ↓ | SSIM ↑ | Whole↑ | Edited↑ |
| **PnP+DDIM Baseline** | | 28.22 | 22.28 | 113.46 | 83.64 | 79.05 | 25.41 | 22.62 |
| *w/* **Pivot** | *w/o* **ZigZag** ($a = 0$) | **19.37** | **25.48** | **77.91** | **50.11** | **83.09** | 24.94 | 22.22 |
| | *w/* **ZigZag** ($a = 0.2$) | 20.06 | 25.29 | 79.94 | 50.99 | 82.91 | 25.00 | 22.33 |
| | *w/* **ZigZag** ($a = 0.6$) | 21.94 | 24.86 | 84.69 | 54.01 | 82.41 | 25.11 | 22.54 |
| | *w/* **ZigZag** ($a = 1$) | 23.46 | 24.55 | 86.10 | 55.04 | 82.18 | **25.43** | **22.91** |
| **Random Pivot** *w/* **ZigZag** ($a = 1$) | | 12.53 | 27.16 | 66.57 | 35.43 | 83.91 | 24.16 | 21.30 |
| **P2P+NTI Baseline** | | 13.44 | 27.03 | 60.67 | 35.86 | 84.11 | 24.75 | 21.86 |
| *w/* **Pivot** | *w/o* **ZigZag** ($a = 0$) | **4.97** | **29.79** | **36.62** | **19.89** | **86.71** | 23.93 | 20.94 |
| | *w/* **ZigZag** ($a = 0.2$) | 5.20 | 29.64 | 37.17 | 20.14 | 86.66 | 23.99 | 21.08 |
| | *w/* **ZigZag** ($a = 0.6$) | 12.51 | 26.71 | 54.94 | 33.05 | 84.98 | 24.85 | 22.01 |
| | *w/* **ZigZag** ($a = 1$) | 16.15 | 25.67 | 84.28 | 49.06 | 82.14 | **25.16** | **22.13** |
| **Random Pivot** *w/* **ZigZag** ($a = 1$) | | 14.72 | 26.29 | 76.71 | 44.47 | 82.72 | 24.44 | 21.43 |

PnP (Tumanyan et al., 2023) *w/* DDIM inversion, Pix2Pix-Zero (Parmar et al., 2023), Instruct-pix2pix (Brooks et al., 2023), and Masactrl (Cao et al., 2023). The improvements are mostly tangible, and we circle some of the subtle discrepancies of the P2P and PnP baselines and the other compared methods in red. Best viewed with zoom in.

## 3 LIMITATIONS AND FUTURE WORK

While our method achieves promising results, it still faces some limitations. For example, our ZZEdit paradigm needs to find a suitable pivot before editing, which takes some time. Generally speaking, on a single Tesla A100 GPU, it takes about 23 seconds for an input image on average. Nevertheless, we argue that it is worthwhile to spend some time for higher editing consistency and background fidelity.

Moreover, we find that GPT-4V (OpenAI, 2023) can act as a good editing evaluator, so we hope to use it to build a new GPT-4V evaluation metric for text-driven image editing in the future.

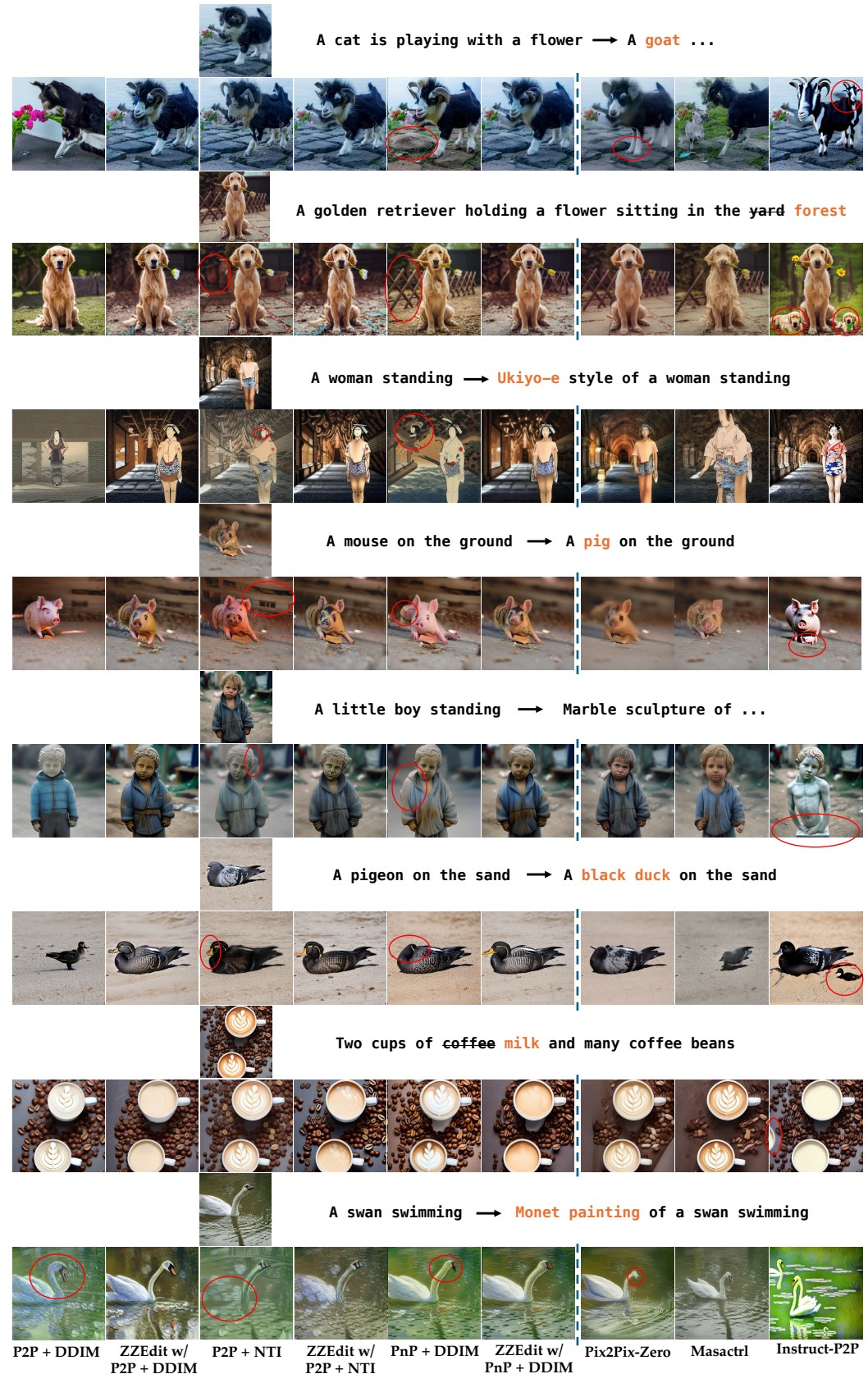

**A-Fig. 2:** More visualization results of different editing techniques.

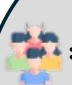 : Given the image and the editing prompt, how to evaluate the following editing results?

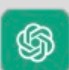 : **Row 1 (Images 2-11)**

- **Early Images (2-5):** These show subtle changes, primarily in color, without altering the bird's physical structure significantly. They do not yet achieve a distinct parrot appearance.
- **Middle Images (6-8):** Here, the changes become more pronounced, especially in terms of color and some slight alterations to the beak and tail. These images start to blend features of both a kingfisher and a parrot.
- **Later Images (9-11):** These images show more advanced transformations where the beak shape, color patterns, and overall morphology are more reminiscent of a parrot. Image 11, in particular, features a complete color transformation with a vivid red and green palette that is very characteristic of many parrot species.

**Row 2 (Images 12-21)**

- **Early Images (12-14):** These continue to build on the transformation, experimenting with different color schemes that align more with various parrot species.
- **Middle Images (15-17):** These edits introduce more dramatic changes to the beak and head structure, aiming for a more realistic parrot look. Image 17 stands out with a well-defined parrot beak and a balanced color scheme that blends naturally.
- **Later Images (18-21):** These images might push the transformation a bit too far, as some edits introduce elements (like the green leaves or overly saturated colors) that detract from the realistic portrayal of a parrot.

## Conclusion

**Image 17** from Row 2 appears to be the best transformation. This image effectively balances the transition into a parrot by incorporating realistic alterations to the beak, head, and body while maintaining natural color blending and textural details. The image achieves a believable transformation from a kingfisher to a parrot, adhering closely to the editing prompt with high visual quality.

**A-Fig.** 3: Using GPT-4V(ision) system (OpenAI, 2023) for evaluating the editing example of Fig. 4 in our main paper. Here, we explore the effect of using different inversion-degree latent as the editing pivot with or without the ZigZag process equipped. We suggest using Fig. 4 as a reference.

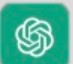 : Given the image and the editing prompt, how to evaluate the following editing results?

: **Row 1 (Images 2-11)**

- **Images 2-5**: Gradual fading of the field into the beach. The earlier images retain more of the field, and by images 4 and 5, the beach starts to become more prominent.
- **Images 6-8**: These images show an effective blend where the beach setting becomes dominant. Image 8, in particular, stands out with a clear, well-integrated beach background that matches the lighting and perspective correctly.
- **Images 9-11**: Full transition to the beach is completed. Image 10 adjusts the position and angle slightly, which might affect the perceived naturalness of the setting.

**Row 2 (Images 12-21)**

- **Images 12-14**: Initiation of the transition with subtle changes in the background. These are more tentative steps towards the beach setting.
- **Images 15-17**: Clear beach backgrounds are established here, with changes in the position and orientation of the woman to better suit the new setting. Image 17 is particularly notable for its realistic integration of the subject with the beach environment.
- **Images 18-21**: These final images explore different angles and further adjustments to the background and the subject's placement. Image 20 offers a dynamic angle with a strong visual composition that aligns well with the beach environment.

**Conclusion**

**Image 17** from Row 2 is the best transformation among all 20 images. This image showcases a fully realized beach setting with a natural and convincing placement of the woman. It achieves a seamless integration by matching the lighting conditions and perspective, ensuring the subject fits organically into the beach environment. The composition is harmonious, and the transition from the original field setting is handled smoothly, making it a standout in fulfilling the editing prompt effectively.

**A-Fig. 4**: Using GPT-4V(ision) system (OpenAI, 2023) for evaluating the editing example of A-Fig. 1 in this Appendix. Here, we explore the effect of using different inversion-degree latent as the editing pivot with or without the ZigZag process equipped. We suggest using A-Fig. 1 as reference.