# OpenReview forum: "ZZEdit: ZigZag Trajectories of Inversion and Denoising for Zero-shot Image Editing"
_ICLR.cc/2025/Conference — ICLR 2025 Conference Withdrawn Submission_

### Official Review · Reviewer_xYZW · 2024-10-28

**Soundness:** 2
**Presentation:** 3
**Contribution:** 2
**Rating:** 5
**Confidence:** 5

**Summary:**

This paper tackles the need for editability and fidelity in text-driven image editing, where edits should match the prompt while preserving other details. Existing methods invert images to Gaussian noise for editing but lose structure. To improve this, ZZEdit introduces a better starting point along the inversion path, using a ZigZag process that alternates denoising and inversion to guide edits while retaining background fidelity. Experiments show ZZEdit’s enhanced effectiveness across various editing tasks compared to current methods.

**Strengths:**

1. The overall presentation is clear and easy to follow.
2. The experimental results seem promising over previous methods.
3. The introduction to the background of the "inversion-and-then-editing" pipeline is clear.
4. The proposed idea seems interesting.

**Weaknesses:**

1. The authors claim the essential of "locating a structure-preserving while sufficient-for-editing point as the editing pivot" and give the illustration in Fig.2. However, the searching process in lines 1-10 (Part I) in Algorithm 1 is not verified. Specifically, why simply using $||\epsilon^{tgt}-\epsilon^{null}||>||\epsilon^{src}-\epsilon^{null}||$ as the condition to judge the pivot timestep? The authors are advised to provide a theoretical analysis to prove it.

2. Similarly, I did not fully understand the potential motivation behind the ZigZag operation (inversion-denoising loops) around the pivot timestep since there is no derivation or supervision signal to regularize. Could the authors provide more detailed illustrations (like mathematical ones) to prove why this operation can handle the aforementioned problem?

3. Something I think that should be corrected is that the "inversion-and-then-editing" pipeline does not have to require achieving the final latent $z_T$, many methods including P2P simply perform inversion to around 0.7T and then denoising.

4. Since the authors have used large language models to evaluate the editing performance and determine the pivot, I advise the authors can add some LLM-related metrics [1] [2] in the comparison evaluation experiments. Also, more inversion methods should be included in the comparison such as [3],[4], etc.

[1] TIFA: Accurate and Interpretable Text-to-Image Faithfulness Evaluation with Question Answering

[2] Diffusion-Model Based Image Editing: A Survey

[3] Edict: Exact diffusion inversion via coupled transformations

[4] Effective real image editing with accelerated iterative diffusion inversion

**Questions:**

Please see the weakness part. I am happy to increase my rating if the concerns get addressed.

---

### Official Review · Reviewer_bXnf · 2024-10-29

**Soundness:** 2
**Presentation:** 2
**Contribution:** 2
**Rating:** 3
**Confidence:** 5

**Summary:**

The paper proposes a novel image editing method that enhances fidelity and editability in text-driven editing pipelines. The approach involves identifying a pivot latent in the inversion process based on specific criteria, using a zigzag step to move from the pivot latent to a more editable latent, and finally performing denoising from this point to achieve the editing goal.

**Strengths:**

1. Introduces the novel ZZEdit method, which aims to improve editability and fidelity in image editing tasks.
2. The idea of encoding each image to a different latent that enhances editability and fidelity for individual cases is promising.

**Weaknesses:**

1. Writing improvements are necessary. The paper’s structure is difficult to follow, and the purpose of the pilot analysis is unclear and doesn’t connect with the method. Additionally, the related work section could better emphasize the distinctions between previous research and the proposed approach to clarify the paper's positioning.
2. The experiments appear insufficient to fully support the proposed claims. Based on the results, it’s challenging to conclude that the proposed method enhances editability and fidelity.

**Questions:**

1. What is the purpose of the pilot experiment? It seems that a mid-inversion latent can secure greater editability than a latent at T steps. Although we observe that one-step denoising in specific latents results in more significant divergence, this alone does not necessarily imply that these particular latents offer better editability. The intermediate latent during DDIM inversion appears to preserve more structure than those fully encoded at T steps. Therefore, can we conclude that these intermediate latents are indeed better for editability?
2. Why do we need to use the null text as the criterion to determine the pivot? And why is this an optimal choice for balancing editability and fidelity?
3. In Figure 4, as the pivot nears T, the zigzag steps decrease. What if the number of zigzag steps were kept consistent across all pivots? Doing so might clarify the effect of zigzag steps at each pivot.
4. In Table 1, it seems a large guidance scale (e.g., 7.5) is used for reconstructing input images with DDIM, whereas a guidance scale of 1 is applied in reconstructions using ZZEdit. If this is the case, it does not seem like a fair comparison. A consistent guidance scale of 1 should be used across methods, or alternatively, baseline methods designed to preserve the reconstruction path, such as NTI, should be used for comparison.
5. As noted in Section 5.3, Table 2 shows that comparing NTI and ZZEdit (w/ NTI) results in decreased fidelity but increased editability. If this is the case, can ZZEdit truly be considered a method that improves both editability and fidelity?

---

### Official Review · Reviewer_pe3M · 2024-10-29

**Soundness:** 2
**Presentation:** 2
**Contribution:** 1
**Rating:** 3
**Confidence:** 3

**Summary:**

This paper is motivated by the idea that reverting an image fully to $x_T$ may not be necessary; instead, a balanced point may exist between editability and preserving the original image's identity. To identify this optimal point, the authors utilize conditional noise prediction. They then perform multiple inversion-sampling steps, referred to as the 'zigzag process.' Finally, they integrate their approach with several existing image editing methods and demonstrate the effectiveness of their work.

**Strengths:**

- The motivation of the paper is clear.
- I appreciate the structure of their approach (i.e., pilot experiment/observation followed by the proposed method).

**Weaknesses:**

- **Results Assessment**
  - The results do not appear promising. For instance, the dog in the main figure does not resemble a Pixar animation style.
  - Additionally, the road lane in ZZEdit with PnP + DDIM is not well-preserved, which the authors identify as a drawback of P2P + DDIM.
  - Furthermore, it is unclear whether this approach can handle cases requiring significant modifications, such as transforming a flamingo into a giraffe.

- **Pilot Experiment and Pivot Point Concerns**
  - The interpretation of the pilot experiment and their method for identifying a suitable pivot point are unconvincing. In my opinion, it’s hard to distinguish between the effects of the text and the inherent characteristics of the diffusion process, from coarse to fine detail. Early steps start with similar noise, making the images naturally alike, and in later steps, the fine details reduce differences again.
  - Even if this concern is addressed, it’s unclear how lines 240-241 relate to their method for identifying the pivot point. To me, this line suggests that the point where the guidance degree reaches its maximum should be the optimal pivot point. If that’s the case, why? Additionally, does the method you use to identify the optimal point—where there’s a stronger response to the target text condition—actually coincide with the point where the guidance degree is maximized? Further explanation would be helpful.

- **Suggestions for Pivot Point Analysis**
  - More analysis on designating pivot points would be helpful. Is the stronger response to the target text condition compared to the source text condition consistently observed beyond the initial point (i.e., the chosen pivot point), or is it stochastic?
  - I’m curious if the results in Figure 6 align with our intuition. For example, were images with a pivot close to 1.0T cases that required severe modification, while images with a pivot at 0.4T were cases where minimal modifications, like color changes, were sufficient? Such an analysis could potentially support the method for determining the pivot point.

- **Zigzag Process Concerns**
  - The motivation or theoretical explanation for the zigzag process is lacking.
  - I can see the clear improvements brought by the zigzag process, but why does it work? Further explanation would be helpful.
  - I’m curious about the results of combining this approach with the editing method used in [1](i.e., editing only through inversion&sampling) . If the guidance from the zigzag process is truly effective, it should work well in conjunction with the approach used in [1].

[1] CFG++: Manifold-constrained Classifier Free Guidance for Diffusion Models

**Questions:**

Please refer to the weaknesses section and typos below.

- Pseudo algorithm line 13. executs > executes
- line 474. steadyly > steadily

---

### Official Review · Reviewer_u3cg · 2024-11-04

**Soundness:** 2
**Presentation:** 3
**Contribution:** 2
**Rating:** 5
**Confidence:** 5

**Summary:**

It proposes a novel editing paradigm dubbed ZZEdit, which first locates such a pivot during the inversion trajectory and then mildly strengthens target guidance via the proposed ZigZag process.

**Strengths:**

It proposes a novel editing paradigm dubbed ZZEdit, which first locates such a pivot during the inversion trajectory and then mildly strengthens target guidance via the proposed ZigZag process.

The writing is good. Very easy to understand.

**Weaknesses:**

1. In line 6 of the algorithm, is there a possibility that, in certain cases, the value of \( P \) could be very small? If so, this would significantly restrict the editing space.

2. In Table 2, there is an unusual phenomenon: when using P2P, the performance of ZZEdit is much worse than PnP-inv. However, when switched to PnP, the performance of ZZEdit suddenly improves drastically, far surpassing PnP-inv. The reviewer requests a detailed and logical explanation for this discrepancy, as well as an in-depth analysis. From an ablation perspective, what is actually driving this improvement—PnP or ZZEdit? If ZZEdit is genuinely effective, why does it perform worse than PnP-inv when using P2P?

3. Almost all experiments focus on ablation studies or comparisons with baselines. The reviewer believes it is more important to include comparisons with state-of-the-art (SOTA) image editing methods, especially given the many training-based methods from 2024 that have already demonstrated mature and effective results. In this context, what is the motivation for introducing ZZEdit, which seems to exhibit only average performance? The reviewer suggests that the authors include direct comparisons with 4 to 5 SOTA methods from 2024, rather than selectively comparing with training-free methods.

4.The conclusions drawn from Table 1 do not strongly support the significance of the zigzag component. When zigzag is set to 0, the consistency with the original image is optimal. However, introducing zigzag causes a significant drop in 80% of the metrics, with only a minor improvement in the CLIP score. These results fail to justify the value of zigzag, as its gains seem minimal while its drawbacks are considerable. Since zigzag is the primary contribution of this work, without it, the paper mainly consists of engineering techniques.

5.The paper only addresses the simplest type of image editing, which is object replacement. It’s worth noting that image editing includes around a dozen types of tasks. For a method that claims to be a general-purpose image editing approach, it should at least demonstrate additional cases, such as object removal, action editing, and background replacement—three very common types of edits. If this method is only capable of object replacement, it should not be presented as a general-purpose editing solution.

**Questions:**

I may change my rating if the authors can give a good response.

---

### Note · Authors · 2024-11-12

I have read and agree with the venue's withdrawal policy on behalf of myself and my co-authors.